

# The impact of different agroecological conditions on the nutritional composition of quinoa seeds

María Reguera[1,2], Carlos Manuel Conesa[1], Alejandro Gil-Gómez[1], Claudia Mónika Haros[3], Miguel Ángel Pérez-Casas[1], Vilbett Briones-Labarca[4,5,6], Luis Bolaños[1], Ildefonso Bonilla[1], Rodrigo Álvarez[5], Katherine Pinto[6], Ángel Mujica[7] and Luisa Bascuñán-Godoy[4,6,8]

[1] Departamento de Biología, Universidad Autónoma de Madrid, Madrid, Spain
[2] Centro de Biotecnología y Genómica de Plantas, Universidad Politécnica de Madrid (UPM)—Instituto Nacional de Investigación y Tecnología Agraria y Alimentaria (INIA), Universidad Politécnica de Madrid, Madrid, Spain
[3] Instituto de Agroquímica y Tecnología de los Alimentos, Paterna, Valencia, Spain
[4] Centro de Estudios Avanzados en Zonas Áridas (CEAZA), La Serena, Chile
[5] Food Engineering Department, Universidad de La Serena, La Serena, Chile
[6] Instituto de Investigación Multidisciplinar en Ciencia y Tecnología, Universidad de La Serena, Chile
[7] Universidad Nacional del Altiplano, Puno, Perú
[8] Laboratorio de Fisiología Vegetal, Departamento de Botánica, Facultad de Ciencias Naturales y Oceanográficas, Universidad de Concepción, Concepción, Chile

Corresponding author
María Reguera,
maria.reguera@uam.es

## ABSTRACT

Quinoa cultivation has been expanded around the world in the last decade and is considered an exceptional crop with the potential of contributing to food security worldwide. The exceptional nutritional value of quinoa seeds relies on their high protein content, their amino acid profile that includes a good balance of essential amino acids, the mineral composition and the presence of antioxidants and other important nutrients such as fiber or vitamins. Although several studies have pointed to the influence of different environmental stresses in certain nutritional components little attention has been paid to the effect of the agroecological context on the nutritional properties of the seeds what may strongly impact on the consumer food's quality. Thus, aiming to evaluate the effect of the agroecological conditions on the nutritional profile of quinoa seeds we analyzed three quinoa cultivars (Salcedo-INIA, Titicaca and Regalona) at different locations (Spain, Peru and Chile). The results revealed that several nutritional parameters such as the amino acid profile, the protein content, the mineral composition and the phytate amount in the seeds depend on the location and cultivar while other parameters such as saponin or fiber were more stable across locations. Our results support the notion that nutritional characteristics of seeds may be determined by seed's origin and further analysis are needed to define the exact mechanisms that control the changes in the seeds nutritional properties.

## INTRODUCTION

*Chenopodium quinoa* Willd., commonly known as quinoa, belongs to the family Amaranthaceae (*Cusack, 1984*). Quinoa is mainly growing in the arid and semi-arid areas of the Andean region of South America although its cultivation has been expanded worldwide (*Choukr-Allah et al., 2016*; *Bazile et al., 2016*). It is well adapted to extreme conditions including water deficits, low temperatures, salinity and poor soils and it can grow at sea level up to elevations of 4,000 m above the sea (*Adolf, Jacobsen & Shabala, 2013*; *Jacobsen, Mujica & Jensen, 2003*; *Jacobsen et al., 2007*; *Ruiz et al., 2014*). In the last years, quinoa has gained worldwide attention due to the remarkable nutritional properties of its seeds, that include high protein content and essential amino acids (including lysine), fats, flavonoids, vitamins and minerals and as a gluten-free product (*Alvarez-Jubete, Arendt & Gallagher, 2010*; *Lutz, Martínez & Martínez, 2013*; *Paśko et al., 2008*; *Gómez-Caravaca et al., 2012*). The seeds, leaves, tender stems and inflorescences can be consumed in the human diet and as animal feed. Also, quinoa leaves are rich in phenolic compounds (including ferulic, sinapinic, gallic acid, kaempferol, isorhamnetin, or rutin) that possess antioxidant and anticancer properties (*Gawlik-Dziki et al., 2013*). Due to the high nutritional value of quinoa, its genetic diversity and its great adaptability to stressful environments, it has been considered an exceptional crop with the potential of contributing to food security worldwide (*Bazile et al., 2016*).

The quality of the seeds and other plant organs is a complex trait that results from the interaction of genetic and environmental factors (*Wimalasekera, 2015*). Breeding programs in quinoa have mainly focused on the generation of better environmentally adapted plants with improved resistance to mildew aiming to develop high-yielding varieties allowing the worldwide crop expansion (*Bazile, Jacobsen & Verniau, 2016*; *Zurita-Silva et al., 2014*). Less attention has been paid to the seed nutritional properties when developing new quinoa varieties. However, the improvement of quinoa seed quality is challenging and key for food security and has been almost exclusively focused on generating hybrid varieties with lower saponin contents (sweet varieties) (*Zurita-Silva et al., 2014*).

The impact of agroecological conditions or agronomic practices on the quinoa nutritional quality has been little explored. Nonetheless, several studies point to the importance of environmental or agronomical factors affecting the nutritional properties (such as the amino acid profile or the protein content) of quinoa seeds including drought, salinity or the cultivation area (*Prado et al., 2014a*; *Gonzalez et al., 2012*; *Bascuñan Godoy et al., 2015*; *Wu et al., 2016*; *Aloisi et al., 2016*). Still, the information regarding the mechanisms that control seed nutritional properties is scarce.

Overall, knowing that the nutritional properties of quinoa seeds are determined by the concentration of nutrients (i.e., amino acid content), their balance and quality (i.e., protein quality or the amino acid profile) and their bioavailability (determined by components such as phytate content), our working hypothesis states that the environmental and climatic factors as well as the agronomical practices used can significantly affect the nutritional quality of quinoa seeds. In order to analyze the impact of these factors, three quinoa cultivars were selected (Salcedo-INIA, Titicaca and Regalona) and their agronomical

performance and nutritional characteristics were evaluated at different locations (Spain, Peru and Chile). The results revealed that several nutritional parameters such as the protein content, the amino acid profile or the mineral composition of the seeds change with the location and cultivar while other parameters such as saponin, remained unchanged among the varieties and locations studied.

## MATERIAL AND METHODS

### Plant material, experimental design and locations

Quinoa (*Chenopodium quinoa* Willd.) seeds of cultivars Regalona (registered variety of BAER, Chile), Salcedo-INIA (experimental station of Illpa—Puno, Peru, *Mujica, Izquierdo & Marathee, 2001*) and Titicaca (generously provided by Dr. Jacobsen of Copenhagen University) were selected to evaluate their agronomic potential and seed nutritional traits at three locations with different agroecological conditions: El Pobo (Teruel, Spain), Arequipa (Peru) and Río Hurtado (Chile). The field experiment at El Pobo (40.50°N, 0.86°W, 1,399 m a.s.l.) (Spain) was carried out under rainfed conditions between May and October 2016 within a range of temperatures between 24.1 and 4.7 °C on average (registering a maximum of 32.3 °C and a minimum of −2.8 °C) and 194 mm total precipitation during the mentioned period. The field trial at Río Hurtado (30.3°S, 70.6°W, 1,462 m a.s.l.) (Chile), was carried out between November 2015 and April 2016 within a range of temperatures between 11 and 25 °C on average (registering a maximum of 34 °C and a minimum of 3.7 °C). The total precipitation during that period was 150 mm and the irrigation was applied using drip lines that released 4 L m$^{-1}$ h$^{-1}$ according to *Martinez et al. (2009)*. In Arequipa (16°S, 71°W, 2,355 m a.s.l.) the experiment was carried out under irrigation between March and July 2016 with an average temperature of 14 °C (registering a maximum of 28.8 °C and a minimum of 4.2 °C) and 15.3 mm total precipitation.

The soil type in Spain (a clay-silty-loam soil) presented a pH of 7.9, 4.8% organic matter, 3 dS m$^{-1}$ of electrical conductivity (EC) of the saturated paste and phosporous (P) and potassium (K) equivalent to 37 and 438 mg kg$^{-1}$, respectively. The soil type in Chile (a loamy alluvial Entisol) presented a pH of 7.8, 7.7% of organic matter, 2.6 dS m$^{-1}$ of EC and content of P and exchangeable K equivalent to 49.96 and 237 mg kg$^{-1}$, respectively. The soil type in Arequipa (Peru) (a loam soil) presented 4.89% of organic matter, 2.25 dS m$^{-1}$ of EC, a pH of 6.95 and a content of 39.31 mg kg$^{-1}$ of P and 624.96 mg kg$^{-1}$ of K.

The experimental design consisted in randomized blocks (8 m$^2$ per block, 4 m × 2 m, L × W) with 4 replications in each location using the three varieties (Regalona, Salcedo-INIA and Titicaca). Each block was composed of 4 rows of 4 m in length (row spacing = 50 cm). Seeds were directly germinated in the soil with a sowing density of 10 kg/ha between 1 and 2.5 cm depth.

### Quantitative multi-elemental analysis

Quantitative multi-elemental analysis by inductively coupled plasma (ICP) spectrometry was used to determine total content of calcium (Ca), magnesium (Mg), iron (Fe), sodium (Na), zinc (Zn), P and K contents in *C. quinoa* seeds. Seed samples were firstly grinded to a fine powder. The samples were then submitted to the Interdepartmental Investigation

Service Laboratory at the Universidad Autónoma de Madrid (SIdI-UAM, Madrid, Spain). Samples were digested in a microwave oven and subsequently analyzed using the equipment ICP-MS NexION 300XX (Perkin Elmer Inc., Hopkinton, MA, USA).

## Total phytate content

Total phytate content was determined using Myo-Inositol Hexakisphosphate Determination method. This method involved acid extraction of myo-inositol phosphates from 0.5 g of flour (ground seeds) in 20 mL of HCl 0.66 M with vigorously stirring at room temperature overnight followed by treatment with a phytase and alkaline phosphatase (K-PHYT 07/11; Megazyme, Bray, Wicklow, Ireland). The total phosphate released was proportional to myo-inositol hexakisphosphate in non-processed seeds. It was measured using a colorimetric method with ammonium molybdate reactive to form 12-molybdophosphoric acid, which was subsequently reduced under acidic conditions to molybdenum blue. The amount of molybdenum blue formed in the reaction was proportional to the amount of free phosphate present in the sample and was measured at 655 nm using a spectrophotometer SPECTROstar nano (BMG LabTech GmbH, Ortenberg, Germany). Phosphorus was quantified interpolating from a calibration curve using standards of known phosphorus concentration. The samples were done by triplicate and the results were expressed in grams of phytic acid per 100 g of seeds in dry matter.

## Protein content

The Kjeldahl method with a conversion factor of 6.25 by AOAC method no 960.52 (*Association of Official Analytical Chemists International, 2016*) was employed to quantify the total crude protein content of the quinoa seed samples. All determinations were done in triplicate.

## Amino acid quantification

Liquid chromatography mass spectrometry (LC/MS) was used to determine free amino acid content of *C. quinoa* seeds. Seed samples were grinded to a fine powder. Free amino acid was extracted as described previously by *Hacham, Avraham & Amir (2002)*. One hundred fifty mg of seed powder was homogenized in 400 µL water:chloroform:methanol (3:5:12 v/v) and centrifuged at 14,000 rpm for 2 min. This step was repeated twice and both supernatants were combined and mixed with 200 µL chloroform and 300 µL of water. The resulting mixture was centrifuged at 14,000 rpm for 2 min. The supernatant (corresponding to the water:methanol phase) was subjected to speed-vac to dry and resuspended in 100 µL miliQ $H_2O$.

Free amino acid extracts were submitted to the Chromatography Laboratory at the SIdI-UAM (Spain) for analysis. Amino acid determination was carried out using HPLC-MS with an Agilent system detector composed by an 1,100 series HPLC coupled to a single 6,120 Quadrupole. For the chromatographic separation, 5 µL were injected in an ACE 5 AQ (250 × 4.6 mm, 5 µm) thermostated column at 30 °C with 0.4 mL/min flow rate and binary gradient elution. The elution was performed in $H_2O$ with 0.1% formic acid (v/v) as eluent "A" and acetonitrile (ACN) with 0.1% formic acid (v/v) as eluent "B". The gradient program was as follows for eluent B: 0 min, 0%; 30 min, 100%; 35 min; 100%; 36 min,

0%; 55 min, 0%. The ionization parameters were as follows: positive atmospheric pressure chemical ionization (APcI+), fragmentor voltage 40 V, capillary voltage 2.0 kV, charging voltage 2.0 kV, nebulizer pressure 20 psig, drying gas flow 5 L/min at 350 °C, vaporizer temperature 250 °C, and corona current 4 $\mu$A. Data was recorded scanning from 50 to 250 Da.

### Ferric reducing antioxidant power (FRAP) assay

The FRAP assay was used to determine the antioxidant capacity of seed samples. The procedure described by *Benzie & Strain (1996)* was used with some modifications. Briefly, 30 $\mu$L sample aliquots were mixed with 90 $\mu$L of distilled water and 900 $\mu$L of freshly prepared FRAP reagent at 37 °C (2.5 mL of a 10 mmol/L 2,4,6-tripyridyl-s-triazine (TPTZ) solution in 40 mmol/L HCl with 2.5 mL of 20 mmol/L $FeCl_3$ and 25 mL of 0.3 mol/ L acetate buffer at a pH of 3.6). The absorbance of the reaction mixture was measured spectrophotometrically (atomic absorption spectroscopy (PinAAcle 900F FL HSN, WinLab32 software; Perkin Elmer Inc., Hopkinton, MA, USA) at 593 nm following incubation at 37 °C for 2 h. FRAP concentration was calculated from a calibration curve obtained by linear regression and the results are expressed in Trolox equivalents (mmol TE $100g^{-1}$). The reference used was the synthetic antioxidant Trolox at a concentration of 100 to 1500 mmol in 80% methanol solution, which was tested under the same conditions.

### Fiber and saponin determination

For fiber determination samples were submitted to the SERBILAC laboratory (Universidad Nacional de San Agustín de Arequipa, Peru). Fiber content was determined following the protocol described in AOAC Methods 2016 (*Association of Official Analytical Chemists International, 2016*). Samples were submitted to the Laboratory of quality control (Universidad Nacional del Altiplano Puno, Peru). Total saponin content was determined spectrophotometrically at 528 nm using an SQ-2802 single beam scanning spectrophotometer (UNICO) (*Lozano et al., 2012*). The concentration of saponin was read off from a standard curve of different concentrations of saponin (Calbiochem, CAS 8047-15-2, Darmstadt, Germany) (from 0.5 to 7.5 mg/ L) dissolved in an aqueous solution.

### Statistical analysis

Results are presented as Mean value ± Standard Deviation or Error. Pairwise comparisons were done by using Student $t$-test at a probability level of 5% ($P < 0.05$). Multiple comparisons were done by one-way analysis of variance (ANOVA) followed by Duncan or Tukey HSD *post-hocs* to analyze the quantitative data at a probability level of 5% ($P < 0.05$). The JMP® (ver.11.0) statistical package (SAS Institute) and the Free Software R were used for the statistical analyses.

## RESULTS

### Variations in agronomical traits of *Chenopodium quinoa* cultivars under different agroecological conditions

Among the three quinoa varieties studied, differences were found in terms of seed yield when consider total yield and seed weight per plant (Table 1). Although a larger total

**Table 1  Agronomical traits of three *Chenopodium quinoa* cultivars growing under different agroecological conditions.** Different letters indicate statistical differences at a $P < 0.05$ (Duncan $t$-test).

| Variety | Location | Yield kg/ha | Seed weight per plant (g) | Harvest index | Plant height (m) | Stem diameter (cm) | Panicle length (cm) | Panicle diameter (cm) | Plant weight (g) | Days to flowering | Days to maturity |
|---------|----------|-------------|---------------------------|---------------|------------------|--------------------|---------------------|------------------------|-------------------|-------------------|------------------|
| Regalona | Chile | 2,402.33 ± 465.28bc | 40.33 ± 13.80ab | 0.42 ± 0.22a | 1.30 ± 0.06ab | 15.33 ± 1.53abc | 17.00 ± 1.00d | 11.00 ± 1.00b | 103.00 ± 18.08ab | 70 | 165 |
| Salcedo | Chile | 2,743.33 ± 80.13b | 36.67 ± 7.02bc | 0.47 ± 0.09a | 0.84 ± 0.09e | 11.33 ± 1.53e | 31.66 ± 3.51b | 7.00 ± 0.00d | 78.00 ± 2.65cd | 100 | 180 |
| Titicaca | Chile | 4,300.33 ± 1841.58a | 42.67 ± 9.71ab | 0.50 ± 0.00a | 1.04 ± 0.09d | 12.00 ± 1.00de | 22.33 ± 2.08c | 7.00 ± 0.00d | 86.00 ± 19.31bc | 50 | 105 |
| Salcedo | Peru | 5,170.00 ± 142.82a | 26.40 ± 1.14c | 0.40 ± 0.00a | 1.36 ± 0.01a | 18.25 ± 0.50a | 35.50 ± 1.29a | 15.60 ± 0.71a | 66.00 ± 2.58d | 65 | 145 |
| Regalona | Spain | 2,606.00 ± 0.00bc | 50.60 ± 9.58a | 0.47 ± 0.04a | 1.25 ± 0.05bc | 14.80 ± 2.28bcd | 20.00 ± 2.55cd | 12.00 ± 1.58b | 107.80 ± 15.06a | 63 | 138 |
| Salcedo | Spain | [a] | [a] | [a] | 1.36 ± 0.06a | 16.80 ± 2.77ab | 36.00 ± 2.55a | 9.00 ± 1.00c | [a] | 92 | 187 |
| Titicaca | Spain | 1,526.00 ± 0.00c | 37.20 ± 4.66abc | 0.42 ± 0.03a | 1.15 ± 0.03c | 12.20 ± 1.30cde | 28.00 ± 2.35b | 7.00 ± 1.00d | 89.60 ± 7.67abc | 51 | 119 |

**Notes.**

[a] Seed losses were detected due to defects related to the timing of maturity, the uniformity of maturity and the drydown of plant at seed maturity. Titicaca and Regalona were unable to grow in Peru at the time the experiment was performed.

yield (Kg/Ha) was achieved by Salcedo-INIA grown in Peru followed by Titicaca grown in Chile, this was due to a larger plant density in these two varieties and not because of a better variety performance as revealed by the seed weight per plant (that was the smallest in Salcedo-INIA from Peru). Also, the harvest index parameter values did not differ among the varieties tested. On the contrary, Titicaca in Spain yield less seeds than Regalona (t-Student, $P < 0.05$) and Regalona showed the largest seed weight per plant among cultivars and locations. Besides, Titicaca and Regalona in Peru were unable to yield seeds and Salcedo-INIA in Spain showed important seed losses due to seed dehiscence detected at harvesting related to defects in the timing of maturity, the uniformity of maturity and the lack drydown of plant at seed maturity.

Most of the morphological traits varied with the location and among cultivars (Table 1). Salcedo-INIA cultivar showed the smallest plants in Chile but the biggest plants in Spain. Salcedo-INIA presented the largest stem diameter and panicle length in Peru and Spain while the stem diameter of Regalona was the largest in Chile. Salcedo-INIA showed the biggest panicle length among varieties in the different locations. Plant weight did not differ among locations but varieties as shown in Table 1.

Regarding the analysis of phenological traits, days to flowering and days to maturity were evaluated. Titicaca showed the shortest time to flowering and to maturity at both locations (Chile and Spain), followed by Regalona. Salcedo-INIA presented the largest time to flowering and maturity (approximately 95 days and 184, respectively) in Spain and Chile while reduced the days to flowering and maturity in Peru.

## Mineral composition and phytate content in *C. quinoa* seeds

Quantitative multi-elemental analysis was performed aiming to assess differences in the seed mineral composition among cultivars and locations due to distinct agroecological conditions. When analyzing the effect of the location in each genotype (Figs. 1A–1C), it was observed that Regalona seeds stored larger amounts of mineral nutrients in Chile with exception of P. Salcedo-INIA showed larger quantities of Mg, Fe, Ca and Zn in Peru, while in Chile presented the largest amount of P and the lowest of Na. Titicaca in Spain had larger amounts of Ca, K, P and Na but less amount of Fe and Zn comparing the same variety grown in Chile.

Differences among varieties were also found in each location (Figs. 1C and 1D). In Chile, Regalona cultivar stored the highest amounts of Ca followed by Salcedo-INIA. Salcedo-INIA presented the largest amounts of Fe and K. Titicaca in Chile showed the lowest contents in Ca, P, Mg and Na compared to the other two cultivars. In Spain, Titicaca showed the highest level of K. Salcedo-INIA grown in Spain showed the highest amounts of P and Mg but smallest of Fe, Ca and Na. All the cultivars grown in Spain or Chile showed similar Zn contents.

Overall, a larger accumulation of Mg and Fe tend to appear in Chile. Zn contents changed with the location but remains unchanged within cultivars. Generally, the type of cultivar and location affects the content of certain minerals, indicating that both factors (variety and location) are determinants of the mineral composition of quinoa seeds.

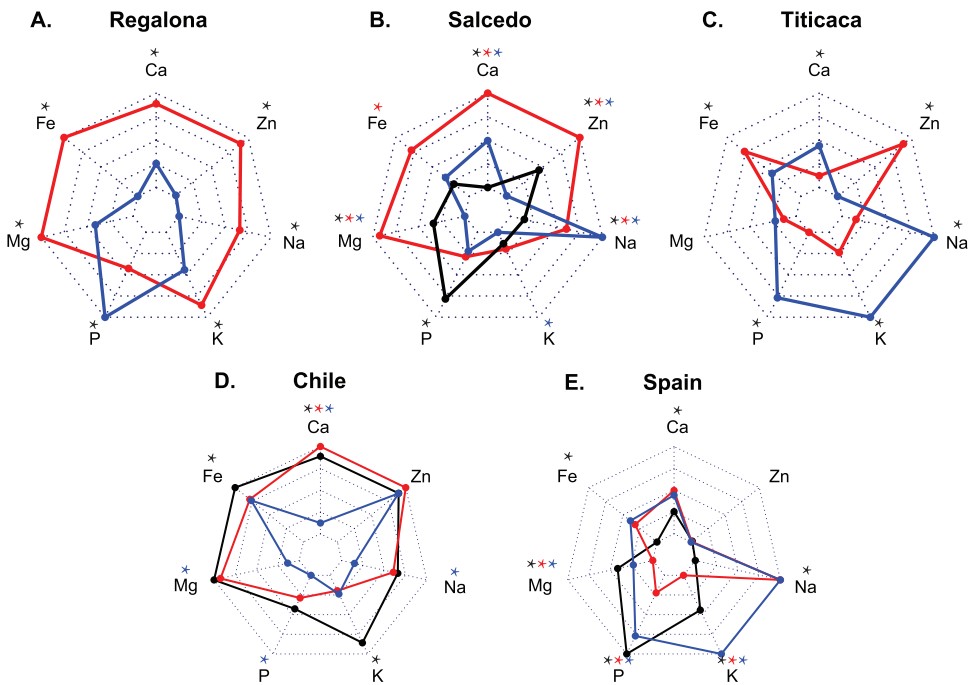

**Figure 1 Mineral composition of three cultivars of *C. quinoa* seeds growing at three different locations.** Quantitative multi-elemental analysis was performed using inductively coupled plasma (ICP) spectrometry and the total content of P, K, Ca, Mg, Fe, Na and Zn were determined. Three cultivars (Regalona, Salcedo-INIA and Titicaca) grown at three different locations (Peru, Chile and Spain) were used to evaluate changes in the mineral composition associated with different agroecological conditions. (A–C) Mineral composition of one variety comparing among countries (Red: Chile; Black: Peru; Blue: Spain): (A) Regalona, (B) Salcedo-INIA and (C) Titicaca. (D–E) Mineral composition of seeds from Spain or Chile comparing among varieties (Red: Regalona; Black: Salcedo-INIA; Blue: Titicaca): (D) Chile and (E) Spain. Values are presented as the Mean relative to the maximum and minimum values for each element ($n = 4$). Asterisks indicate statistical differences at a $P < 0.05$ (Tukey $t$-test or Student $t$-Test when comparing pairs). When more than two samples are compared, colored asterisks indicate the sample that is statistically significant.

Phytic acid is considered an important seed component conditioning the nutritional properties of seeds (*Lott et al., 2000*). The analysis of phytate content in quinoa seeds (Fig. 2) revealed that Regalona seeds from Spain showed the largest phytate content, followed by the Salcedo-INIA and Titicaca seeds from this country. Titicaca seeds from Chile presented the smallest phytate content. The fact that no differences were found among cultivars in the same location but a given genotype vary among locations suggests that phytic acid content in quinoa seeds might be determined by environmental factors and not by the type of cultivar.

## Total Protein content and free amino acid profile of quinoa seeds obtained from Chile, Peru and Spain

The range of total protein content was found between 14 and 17% among the cultivars and the locations analyzed (Fig. 3). No differences were found when comparing among varieties in a particular location. Consistently, seeds from Chile showed a higher total protein content

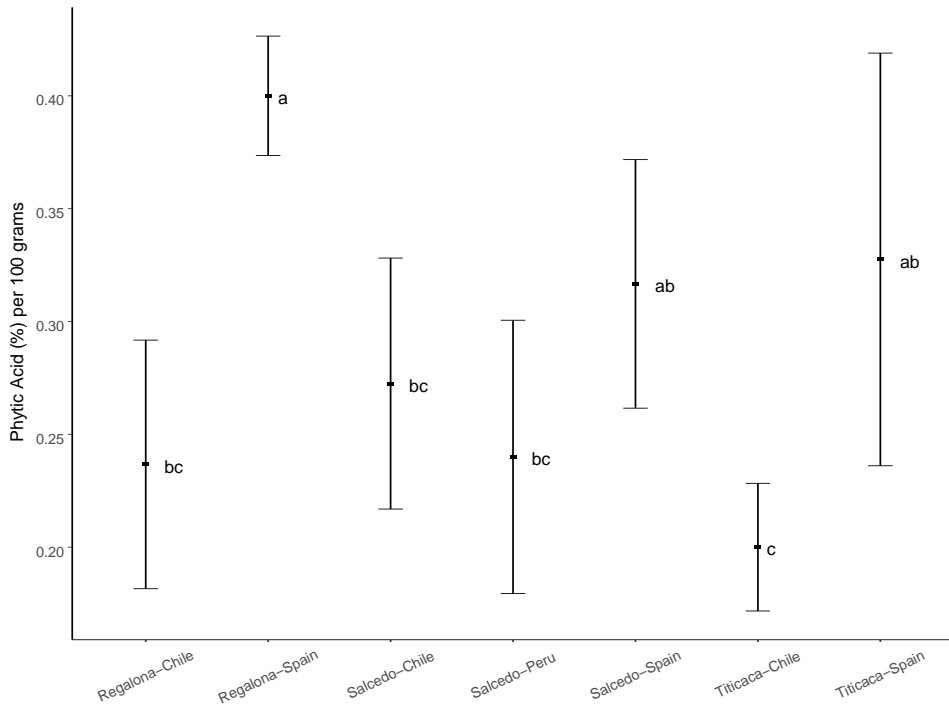

**Figure 2 Phytic acid content among three varieties of *C. quinoa* seeds growing at different locations.**
Phytic acid content of quinoa seeds was determined using the myo-inositol hexakisphosphate method.
Phytic acid is presented as the % of phytate per 100 grams of seeds. Values are Mean ± Stnd. Dev. ($n = 4$).
Different letters indicate statistical differences at a $P < 0.05$ (Duncan $t$-test).

comparing to Spain or Peru. These results might suggest that agroecological conditions can influence on the protein content of the seeds.

Regarding amino acid composition, the most abundant amino acids found in the quinoa seeds analyzed in the present study were arginine and glutamic acid (Fig. 4 and Table S2). Asparagine, glutamic acid, glutamine, histidine, glycine, hydroxyproline, serine and threonine and remained unchanged in all cultivars and locations. The rest of the amino acids quantified showed differences among cultivars and/or locations. For instance, Regalona seeds grown in Spain showed higher contents of arginine, aspartic acid, lysine and methionine compared to the Chilean Regalona seeds. Spanish Salcedo-INIA seeds showed larger contents of aspartic acid, isoleucine, leucine and valine compared to the Peruvian and Chilean Salcedo-INIA seeds. Arginine and phenylalanine showed higher contents in Salcedo-INIA seeds obtained from Spain when compared with the seeds from Chile but no differences were detected in the contents of these two amino acids between Spain and Peru.

Noteworthy, an elevated amount of tryptophan was found in Salcedo-INIA from Peru, amount that was superior to any of the cultivars analyzed. In Spain, the amount of lysine in Regalona and valine in Salcedo-INIA were higher when compared to other cultivars or locations. In contrast, the content of amino acids in Titicaca did not change significantly among cultivars nor locations.

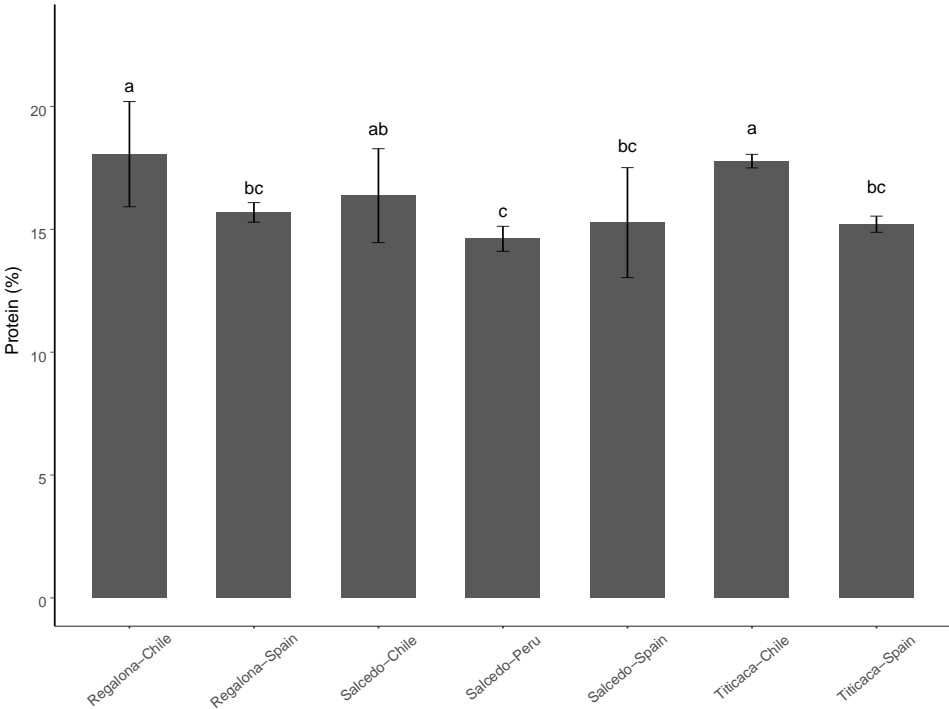

**Figure 3 Total Protein content of *C. quinoa* seeds.** Total protein content was determined by using the Kjeldahl method in seeds of Regalona, Salcedo-INIA and Titicaca cultivars grown in Chile, Peru or Spain. Values are Mean ± Stnd. Dev. ($n = 3$). Different letters indicate statistical differences at a $P < 0.05$ (Duncan $t$-test).

When analyzing the amino acid content by location it was observed that the varieties grown in Chile did not present differences in their amino acid content. However, inter cultivar differences were observed in the amino acid contents in the Spanish quinoa seeds for alanine, asparagine, isoleucine, lysine and valine.

## Antioxidant properties of quinoa seeds in different cultivars and locations

Antioxidant capacity varied among cultivars and locations ranging from 0.75 to 0.15 mmol TE 100 g$^{-1}$ (Fig. 5). Titicaca and Regalona seeds from Chile presented the highest antioxidant activity measured as FRAP, while Salcedo-INIA from Peru presented the lowest values. The big differences observed between the antioxidant properties of Regalona at the two locations studied did not appear in Titicaca nor Salcedo-INIA (differences among locations in these two cultivars were smaller) what would indicate that changes among cultivars appear when evaluating the effect of the agroecological conditions on the antioxidant properties.

## Fiber and saponin contents

Although no differences were found in the saponin content (Fig. 6A) among the cultivars or locations, Titicaca seeds from Chile showed larger fiber contents compared to Regalona seeds obtained from Spain (Fig. 6B). These results might indicate that fiber might be more

Peer J

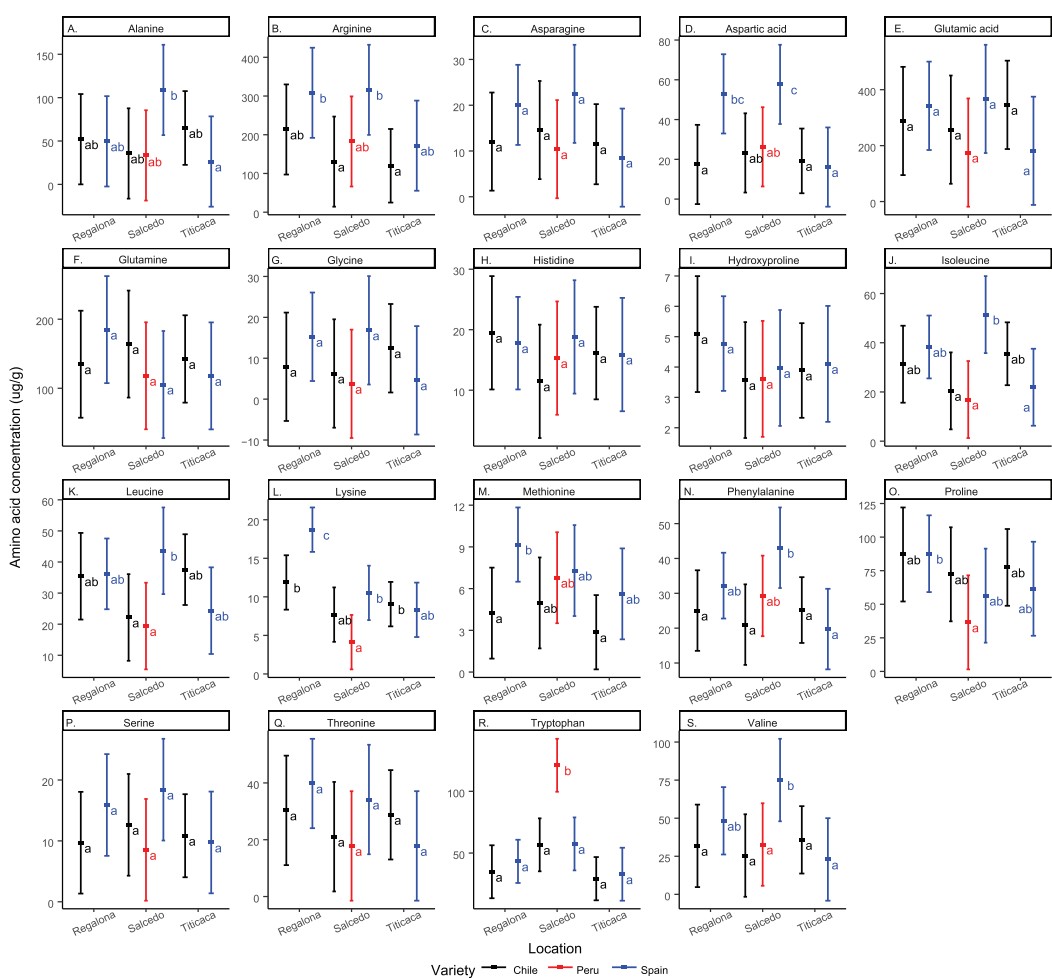

**Figure 4** **Free amino acid composition of *C. quinoa* seeds from different cultivars and locations.**
Amino acid contents of a pool of 10 g of *C. quinoa* seeds were determined by using LC/MS. Seeds of different cultivars including Regalona, Salcedo-INIA and Titicaca were obtained from different locations (Chile, Peru or Spain). (A) Alanine, (B) arginine, (C) asparagine, (D) aspartic acid, (E) glutamic acid, (F) glutamine, (G) glycine, (H) histidine, (I) hydroxyproline, (J) isoleucine, (K) leucine, (L) lysine, (M) methionine, (N) phenylalanine, (O) proline, (P) serine, (Q) threonine, (R) tryptophan and (S) valine. Values are Mean ± SE ($n = 4$). Different letters indicate statistical differences at a $P < 0.05$ (Tukey $t$-test).

susceptible to variations associated with changes in agroecological conditions despite no big differences were found among cultivars and locations.

## DISCUSSION

Quinoa is able to grow under a wide variety of environmental conditions and to tolerate a broad range of stresses which, in addition to the excellent nutritional properties of its seeds, makes this crop an attractive and feasible option from an agronomic perspective (*Jancurová, Minarovičová & Dandár, 2009*; *Filho et al., 2017*). Numerous studies have been published in the last years describing the effect of different abiotic stresses on quinoa, however, the analysis of how quinoa responds to a certain environment altering the nutritional properties

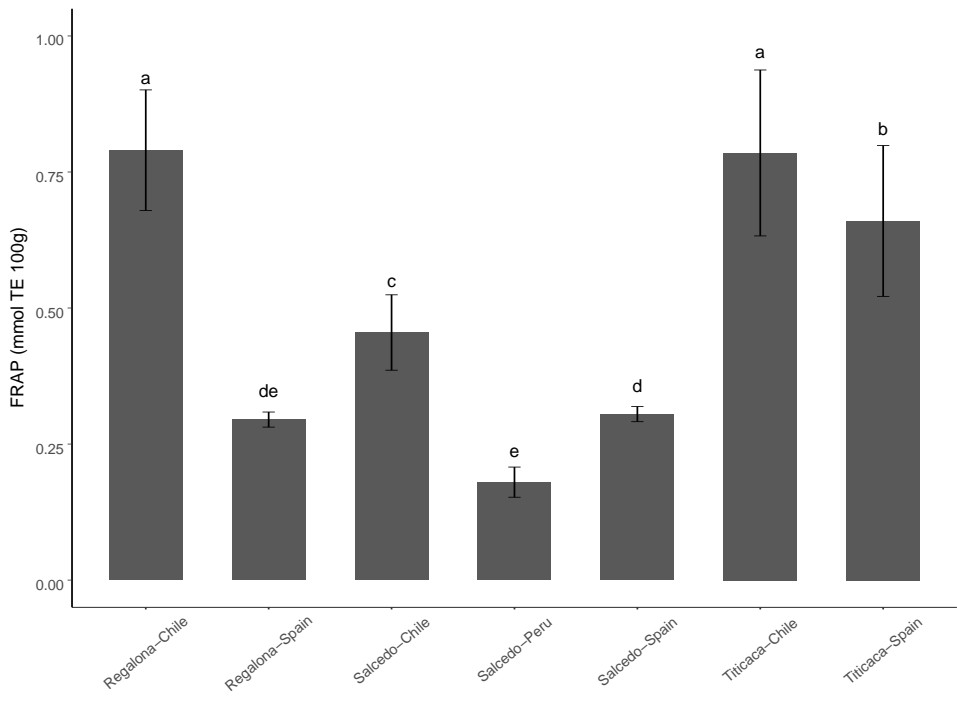

**Figure 5 Variation of antioxidant properties of *C. quinoa* seeds of different cultivars and locations.** Antioxidant properties were determined by using FRAP in seeds of Regalona, Salcedo-INIA and Titicaca cultivars grown in Chile, Peru or Spain. Values are Mean ± Stnd. Dev. ($n = 3$). Different letters indicate statistical differences at a $P < 0.05$ (Duncan $t$-test).

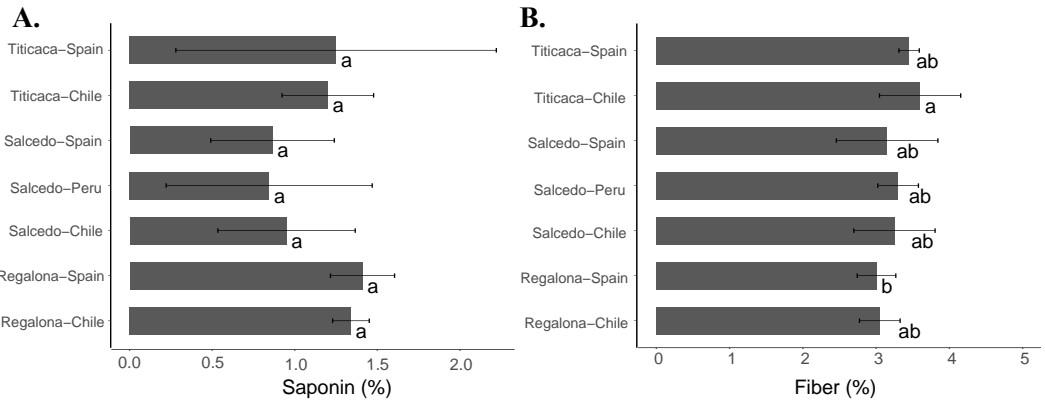

**Figure 6 Saponin and fiber contents of *C. quinoa* seeds from three cultivars grown at three different locations.** (A) Saponin and (B) fiber contents were determined in Regalona, Salcedo-INIA and Titicaca seeds obtained at three different locations (Chile, Peru or Spain). Values are Mean ± Stnd. Dev. ($n = 3$). Different letters indicate statistical differences at a $P < 0.05$ (Duncan test).

of the seeds has been little explored. Here, we have analyzed how different agroecological conditions can induce distinct agronomical and nutritional responses depending on the variety of quinoa considered. We observed that despite some traits were largely influenced by the genotype, such as the phenological characters (*Bertero et al., 2004*), others were more sensitive to the interaction between the genotype and the environment resulting in specific responses.

The analysis of the mineral composition is crucial when considering the nutritional quality of the seeds (*Vaz Patto et al., 2015*). The mineral concentration of the quinoa seeds evaluated in the present study was found within a range similar to what was previously reported in quinoa (Table S1) (*Prado et al., 2014b*; *Miranda et al., 2013b*; *Miranda et al., 2013a*). Interestingly, the accumulation of some minerals was heavily influenced by the location considered. For instance, the amount of Ca varied up to 2.6 times among locations in Salcedo-INIA cultivar. These variations could significantly impact on the consumer. For instance, the dietary reference intake (DRI) for Fe in women between 19 and 30 years old stablished by the US government (*U.S. Department of Health and Human Services, U.S. Department of Agriculture, 2015*) corresponds to 18 mg/day which could be covered with the uptake of 200 g of the Regalona seeds from Chile instead of the 400 g required when consuming Regalona seeds from Spain. The differences observed in the mineral composition may be caused by a strong effect of the environment, and we hypothesized that one of the main contributing factors could be the soil composition. Previous studies growing quinoa under different agroecological conditions are in agreement with our results and hypothesis (*Prado et al., 2014b*; *Miranda et al., 2013a*), and can be extended as well to the effects observed on the grains and seeds of other crop species such as wheat, pea and corn (*Kotlarz et al., 2011*; *Gomez-Becerra et al., 2010*; *Gu et al., 2015*).

One of the most attractive features of quinoa is the high protein content and the good balance of amino acids in the seeds (*Filho et al., 2017*). Both were shown to rely on nitrogen metabolism which is known to be regulated by agroecological factors such as abiotic stresses (i.e., water stress) or soil factors (i.e., nitrogen fertilization) (*Bascuñan Godoy et al., 2015*; *Geren, 2015*; *Thanapornpoonpong et al., 2008*; *Varisi et al., 2008*). Although the total protein content was found within a range already described for quinoa seeds (between 15 and 20%) (*Jancurová, Minarovičová & Dandár, 2009*; *Miranda et al., 2013b*), our results suggest that the environmental conditions could determine the protein content found in quinoa. Similarly, Gonzalez and coworkers found changes in the protein content of ten quinoa cultivars growing in two different agroecological regions (*Gonzalez et al., 2012*). Other studies, however, have shown no differences in the protein content when comparing quinoa seeds growing in two environments (*Miranda et al., 2013a*), suggesting that the interaction of environmental conditions and genotype plays an important role modulating the protein content in the seeds.

The analysis of the seed free amino acid profile revealed also variations associated with the cultivar and location. The presence of essential amino acids (EAA), including methionine, threonine or lysine, contributes to the high nutritional properties of the quinoa seeds and can vary when plants are subjected to abiotic stress (*Bascuñan Godoy et al., 2015*; *Joshi et al., 2010*). Among the EAA analyzed in this study, only threonine and

histidine remained stable in all cases, and changes were observed in the rest of EAA. These results support previous findings that claimed that the EAA content can vary significantly depending on the genotype and seed's origin (*Gonzalez et al., 2012*). Thanapornpoonpong and coworkers found that nitrogen availability determines not only the protein content but also the amino acid composition of quinoa seeds (*Thanapornpoonpong et al., 2008*). Taken all together, these findings suggest that a complex genotype ×environment interaction alters nitrogen metabolism resulting in seed nutritional differences. Nonetheless, the main and specific factors contributing to these changes remain to be elucidated.

Different bioactive components contribute to the antioxidant capacity of quinoa seeds including polyphenols, flavonoids and vitamins (A, B and E) (*Filho et al., 2017*), which may prevent cancer, cardiovascular and other, chronic diseases (*Tang & Tsao, 2017*). The amount of these phytochemicals in quinoa seems to be genotype-specific and could vary significantly under stressful conditions (*Bascuñan Godoy et al., 2015*; *Aloisi et al., 2016*). Our results showed differences among cultivars and locations in agreement with the results obtained by Miranda and co-workers (*Miranda et al., 2013a*), what highlights the importance of the environmental factors conditioning the accumulation of antioxidants in the seeds. In the present study, was especially noticeable the antioxidant capacity of the Chilean cultivars that tended to have the greatest values when comparing among locations.

The total fiber content in quinoa seeds was found within the characteristic range (*Filho et al., 2017*). Little variation was observed among cultivars and locations suggesting that this parameter might be less sensitive to agroecological variations. Nonetheless, Miranda and coworkers reported that the changes in fiber contents only occurred in the soluble dietary fraction and no alteration was detected in the total fiber, indicating that both fractions might be affected differently (*Miranda et al., 2013a*).

Saponin and phytic acid have been traditionally considered antinutrients that diminish the nutritional value of seeds due to their ability to alter the absorption of minerals (*Jancurová, Minarovičová & Dandár, 2009*; *Ruales & Nair, 1993*). However, several studies have pointed to the beneficial effects associated to these two compounds (*Yao et al., 2014*). In the case of saponin different breeding programs have aimed to develop quinoa varieties with a lower saponin content trying to increase the palatability of the seeds increasing consumers acceptance (*Zurita-Silva et al., 2014*; *Nowak, Du & Charrondière, 2016*). Besides being determined by the variety, recent evidences have suggested that external factors might impact on saponin contents of quinoa seeds. For instance, it was reported that the saponin content of Q52 variety diminished under water or salinity stress (*Gómez-Caravaca et al., 2012*). Also, it was shown that the saponin content was altered in Regalona seeds when growing at different locations (*Miranda et al., 2013a*). Nonetheless, our results did not find differences in the saponin content in any of the cultivars or locations analyzed supporting the hypothesis that claims that this trait is largely determined by the variety more than being environmentally regulated.

A different response was observed regarding phytic acid whose content varied with the location suggesting that this might modulate its accumulation. To our knowledge, no previous studies have been carried out evaluating the effect of environmental factors in quinoa seed phytate composition. In oat, barley and dry beans was described a strong effect

of agroecological conditions in their phytate content (*Miller, Youngs & Oplinger, 1980*; *Dai et al., 2007*; *Wang et al., 2017*). However, the exact factors that cause these changes remain unclear. Considering that phytic acid contributes extensively to the nutritional profile of quinoa seeds it should be stressed that further studies are needed in the field to deeply analyze the contributing environmental factors involved in phytate seed accumulation.

## CONCLUSIONS

Our results highlight that different agroecological conditions could significantly alter the agronomical and nutritional properties of quinoa what impacts on the seed quality. Although not all the nutritional traits evaluated varied to the same extent, one can affirm that both the variety and location determined the mineral composition, the amino acid profile, the protein content and the antioxidant capacity of the quinoa seeds. Therefore, when evaluating the nutritional quality of quinoa seeds and in order to provide precise nutritional information to the consumer we should consider the cultivar and the agroecological context. This work also provides valuable information that could be used in breeding programs to maximize the potential of this crop by defining stable varieties and/or environments from a nutritional point of view.

## ACKNOWLEDGEMENTS

We thank Dr. Sven Jacobsen from the University of Copenhagen for providing us with the Titicaca *Chenopodium quinoa* seeds used in this study. We also thank Karla Miranda Ramos for the technical assistance with the phytic acid determination, Rosa Sedano Pérez for her technical assistance with the amino acid determination, Inmaculada Rivas Ramírez for her technical assistance with the minerals quantification and Susana Vilariño for the stimulating discussions that have helped improving this manuscript.

### Funding

This work was supported by the CEAL-AL/2015-27 Banco Santander-UAM grant (Spain), the PROMETEO/2017/189 grant from the Generalitat Valenciana (Spain) and the Juan de la Cierva Fellowship Program (JCI-2012-14172) (MINECO, Spain) (to María Reguera). The funders had no role in study design, data collection and analysis, decision to publish, or preparation of the manuscript.

### Grant Disclosures

The following grant information was disclosed by the authors:
Banco Santander-UAM grant: EAL-AL/2015-27.
Generalitat Valenciana: PROMETEO/2017/189.
Juan de la Cierva Fellowship Program: JCI-2012-14172.

### Competing Interests

The authors declare there are no competing interests.

## Author Contributions

- María Reguera conceived and designed the experiments, performed the experiments, analyzed the data, contributed reagents/materials/analysis tools, prepared figures and/or tables, authored or reviewed drafts of the paper, approved the final draft.
- Carlos Manuel Conesa conceived and designed the experiments, authored or reviewed drafts of the paper, approved the final draft.
- Alejandro Gil-Gómez analyzed the data, prepared figures and/or tables, authored or reviewed drafts of the paper, approved the final draft.
- Claudia Mónika Haros and Vilbett Briones-Labarca performed the experiments, analyzed the data, contributed reagents/materials/analysis tools, authored or reviewed drafts of the paper, approved the final draft.
- Miguel Ángel Pérez-Casas, Rodrigo Álvarez and Katherine Pinto performed the experiments, authored or reviewed drafts of the paper, approved the final draft.
- Luis Bolaños and Luisa Bascuñán-Godoy conceived and designed the experiments, contributed reagents/materials/analysis tools, authored or reviewed drafts of the paper, approved the final draft.
- Ildefonso Bonilla conceived and designed the experiments, contributed reagents/materials/analysis tools, authored or reviewed drafts of the paper, approved the final draft, read critically the manuscript.
- Ángel Mujica conceived and designed the experiments, performed the experiments, contributed reagents/materials/analysis tools, authored or reviewed drafts of the paper, approved the final draft.

## Data Availability

The raw data are provided in the Supplemental Files.

## Supplemental Information

Supplemental information for this article can be found online at http://dx.doi.org/10.7717/peerj.4442#supplemental-information.

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
