# Peer review of "The impact of different agroecological conditions on the nutritional composition of quinoa seeds"

_PeerJ, doi:10.7717/peerj.4442_

## Round 0.1 · original submission · Minor Revisions

Please follow all the suggestion of the reviewers and make sure that references are in accordance with the style of the journal.

Reviewer 1 ·

Basic reporting

No comment

Experimental design

It is correct and well designed to achieve the proposed objectives that are clear. The methods are well explained.

Validity of the findings

The data are robust and correctly analyzed. The conclusions are well stated and provides an answer to the original research question.

Additional comments

The main objective of the present article is to evaluate the influence of the variety and the agroecological conditions on the nutritional properties of quinoa seeds. Three varieties were cultivated in three different countries and the analysis of minerals, phytic acid, saponins, protein, amino acids, fiber, and antioxidant capacity was carried out. The study is very interesting, exhaustive and complete. Also, it is very timely since quinoa production is currently raising worldwide and new studies showing the impact of the environmental and agronomic conditions on the nutritional value of this pseudocereal are required. However, there are some aspects that should be checked and addressed to have an article suitable to be published in the journal. These points are the followings:

1. Materials and Methods:
1.1. Section on “Plant material, experimental design and locations”: the authors should indicate the maximum and minimum temperatures registered in Arequipa region during the study period as they have been indicated for other two regions (line 98). Similarly, the authors should expressed the information provided about the soil type of three different countries in a similar way (units of minerals, pH,…) (lines 99-104).
1.2. The authors should check the results. When decimals are indicated, the comma should be substituted by a period. As an example, in line 99, “4,8” should be substituted by “4.8”.
1.3. The authors should follow the authors’ guidelines to express the units since they use different systems.
1.4. The authors should define the minerals the first time they are cited, and then, they should use the corresponding abbreviation.
1.5. Line 116: The authors should provide the information of Perkin-Elmer.
1.6. Line 127: The authors should indicate the equipment used to measure the absorbance with information of the company that supplied it.
1.7. Line 131: The authors use 6.25 as conversion factor for the Kjeldahl method. Is not a more suitable conversion factor defined for this plant?
1.8. Line 148: Define ACN the first time it is cited.
1.9. Line 157 and 158: Define FRAP and TPTZ (in this case, the full name should be written before the abbreviation).
1.10. Line 163: The authors indicate that the units were expressed as mmmol TE g-1 tomato pulp. I suppose that this is an error and quinoa seed is the correct word.
1.11. Line 168: Provide information of Calbiochem.
1.12. How were the saponins extracted to be quantified?

2. Results:
2.1. Lines 204-207: The sentence should be rewritten for a better understanding.
2.2. Lines 233 and following: The authors should use only the full name or the three-letter code to name the amino acids but not both of them.
2.3. Line 256: The authors should indicate the units used to express the antioxidant capacity (0.75 to 0.15).

3. Discussion:
3.1. Line 284: “2,6” should be substituted by “2.6”.
3.2. Lines 297-302: How can the authors explain the differences found between their results and those results published in the literature? A possible explanation should be included.

4. References: The format should be carefully checked. In addition, the latin names should be written in italics.

5. Table 1:
5.1. Why only one variety (Salcedo) was cultivated in Peru? Was there any problem with the other two varieties?
5.2. Where are the two asterisks in Table 1?

6. Figure 2: As the SD were so big, it is difficult to see the differences in the phytic acid content among varieties but the authors indicate that there were no differences.

7. Figure 4: The figure is a little bit confused because many small figures have been included. Probably, it would be better if only 2 or 3 representative figures are included. In the case that the authors want to show the results of all amino acids, a table would be the best and the clearest manner to show them.

8. As figures 6 and 7 are explain together at the same section, they should be joined in one figure 6A and 6B.

Reviewer 2 ·

Basic reporting

The manuscript is written in a clear language. Enough references are included and the manuscript provide a clear general and direct background.

The structure of the manuscript is suitable. Figures (from 1-7), Tables (Table 1) and Supplementary tables (Supp. Table 1) are relevant and in general appropriately described and labeled. However, minor corrections must be done.

For example:
1.- In Table 1 (**) indicate that plants did not yield seeds. Data for varieties Regalona and Titicaca must be included (even if they are filled only with two asterisks (**).
2.- Figure 1 (figure 1a) must be edited because images are overlapped, consequently labels and asterisks for Na can’t be seen.
3.- Figure 4 must be edited. In the first amino acid mini-graphic (Alanine) there are 8 labels instead of 7.
4.- In supplemental material some raw data is written in Spanish, and units (like ppm or mg/kg) are missing in some of them.


A few typos must also be corrected throughout the manuscript:
Line 169: must be in English
Line 180: consider total yield
Line 188: dehiscense
Line 234: histidine (His) “is duplicated”
Line 237: arginine (Arg)
Line 239: aspartic acid (Asp)
Line 240: Arginine (Arg) and
Line 266: obtained from Spain (Figure 7).
Line 305: when plants are subjected to
Lie 357: point of view.
Line 373: D. Cusack,
Line 407: Świeca, Sulkowski, Czyż
Line 446: FRAP assay,
Line 517: acid concentration,

Methods are described with necessary information to be reproduced.
Minor corrections must be done. For Example:
1.- lines 99-104, don´t use indistinctly ppm, mg/kg mg.kg-1. Choose one.
2.- In all material and methods section use L as a symbol for litter (e.g. µL, mL, L) don’t use uppercase and lowercase letter interchangeably.

Experimental design

no comment

Validity of the findings

no comment

---

## Round 0.2 · accepted · Accept

The manuscript is Accepted, but there are a couple of issues which should be addressed while in production:

Line 271 and Line 273 , “Figure” need to be labelled in the same style of the other figures (according to the Journal style).

“Supplementary Table 2” is not recalled within the main article text.